# Characterizing Land Use Impacts on Channel Geomorphology and Streambed Sedimentological Characteristics

**Sean J. Zeiger** [1,*] **and Jason A. Hubbart** [2,3] 

1   School of Natural Resources, University of Missouri, 203-T ABNR Building, Columbia, MO 65211, USA
2   Institute of Water Security and Science, West Virginia University, 3109 Agricultural Sciences Building, Morgantown, WV 26506, USA; Jason.Hubbart@mail.wvu.edu
3   Davis College of Agriculture, Natural Resources and Design, Schools of Agriculture and Food, and Natural Resources, West Virginia University, 3109 Agricultural Sciences Building, Morgantown, WV 26506, USA
*   Correspondence: ZeigerS@Missouri.edu; Tel.: +1-417-664-5321

**Abstract:** Land use can radically degrade stream physical habitat via alterations to channel geomorphology and sedimentological characteristics. However, independent *and* combined influences such as those of agricultural and urban land use practices on channel geomorphology and substrate composition remain poorly understood. To further understanding of mixed land use influence on stream physical habitat, an intensive, 56 km hydrogeomorphological assessment was undertaken in a representative mixed land use watershed located in Midwestern USA. Sub-objectives included quantitative characterization of (1) channel geomorphology, (2) substrate frequency and embeddedness, and (3) relationships between land use, channel geomorphology, and substrate frequency and embeddedness. Channel geomorphology, and stream substrate data were directly measured at survey transects (n = 561) every 100 m of the entire 56 km distance of the reference stream. Observed data were averaged within five sub-basins (Sites #1 to #5) nested across an agricultural-urban land use gradient. Multiple regression results showed agricultural and urban land use explained nearly all of the variance in average width to depth ratios ($R^2 = 0.960$; p = 0.020; n = 5), and maximum bank angle ($R^2 = 0.896$; p = 0.052; n = 5). Streambed substrate samples of pools indicated significantly (p < 0.001) increased substrate embeddedness at agricultural Site #1 (80%) located in the headwaters and urban Site #5 (79%) located in the lower reaches compared to rural-urban Sites #2 to #4 (39 to 57%) located in the mid-reaches of the study stream. Streambed substrate embeddedness samples of riffles that ranged from 51 to 72% at Sites #1 and #5, and 27 to 46% at Sites #2 to #4 were significantly different between sites (p = 0.013). Percent embeddedness increased with downstream distance by 5% km$^{-1}$ with the lower urban reaches indicating symptoms of urban stream syndrome linked to degraded riffle habitat. Collectively, observed alterations to channel morphology and substrate composition point to land use alterations to channel geomorphology metrics correlated with increased substrate embeddedness outside of mid-reaches where bedrock channel constraints accounted for less than 3% of substrate frequency. Results from this study show how a hydrogeomorphological assessment can help elucidate casual factors, target critical source areas, and thus, guide regional stream restoration efforts of mixed-land-use watersheds.

**Keywords:** physical habitat; aquatic ecology; stream health; environmental flows; land use; hydrology; hydroecology; ecohydrology

## 1. Introduction

The importance of channel geomorphology and stream substrate composition with regards to stream habitat has been well-documented [1–5]. Previous studies indicated that channel geomorphology and stream substrate are important aspects of stream physical habitat [6–8]. For example, substrate is a medium for settlement by propagules [9]. Successful propagules that mature into aquatic plants provide substrate stability, food, dissolved oxygen, physical habitat, and refugia beneficial for healthy stream ecosystems [7]. There is therefore also a direct link between suitable stream substrate composition and the abundance and distribution of macroinvertebrates [8,10], mussels [6], and fishes [11]. Given the influence on stream health, it is important to understand the dominant physical processes that control channel geomorphology and substrate composition.

Process-based understanding indicates channel geomorphology and stream substrate composition is dependent on sediment supply, and transport [12]. Surface runoff and streamflow are largely the erosive forces and transport mechanisms for overland soil erosion and stream bed and bank erosion, respectively [12]. However, a myriad of independent and interacting natural and anthropogenic factors influence surface runoff, streamflow, sediment supply and transport (e.g., meteorological conditions, hydrology, topography, soils and underlying geology, land cover, and human activities) [12]. Thus, regions associated with physiographic characteristics known to increase sediment supply and transport are areas of concern susceptible to alterations of channel geomorphology and stream substrate composition, and therefore physical habitat.

Streams of the Midwestern USA are commonly associated with increased sediment supply and degraded stream health [13]. For example, Gellis et al. [13] found channel sources of sediment accounted for the majority (>50%) of bed sediment in 79% of 99 Midwestern watersheds sampled. Claypan soils overlain by loess are especially subject to increased surface runoff and sediment supply to nearby streams [14,15]. A study by Willett et al. [15] indicated bank sediment accounted for 79% to 96% of the total in-stream sediment in two claypan watersheds of the central USA. A study by Huang [16] in mid-Missouri, showed bank erosion contributed approximately 67% of in-stream suspended sediment load in Hinkson Creek Watershed located in central claypan region of the USA. Given previous findings, it is not surprising that Midwestern streams are commonly characterized by increased frequency of fine bed sediment textures and increased substrate embeddedness related to degraded algal, macroinvertebrate, and fish communities [17–19]. Increased substrate embeddedness can influence stream community composition by reducing available riffle-pool habitat when increased deposition of fine sediments and sand fill interstitial spaces of course gravel substrate [20,21]. In Midwestern streams and other regions associated with increased sediment supply, human activity (i.e., land use) can exacerbate problems and further reduce suitable stream physical habitat and aquatic refugia.

Previous studies showed agricultural land use activities can increase sediment supply potentially leading to problems with sedimentation and substrate embeddedness [13,14,22,23]. For example, conventional tillage operations and livestock presence have been shown to increase erosion rates and sediment supply to nearby stream networks [12]. Given stream access to riparian areas, livestock can trample biomass and erode stream banks thereby increasing sediment supply [22]. Near channel removal of streambank stabilizing riparian vegetation has also been shown to increase bank erosion and alter channel geomorphology in agricultural watersheds [23].

Urban land use has also been shown to increase stream sediment supply. A literature review by Walsh et al. [24] presented consistent confirmation of urban land use influence on watershed hydrology, water chemistry, channel geomorphology, organic matter, algae, macroinvertebrates, fishes, and ecosystem processes. The term 'urban stream syndrome' is widely accepted to indicate consistent urban land use influence on stream health [24,25]. More specifically, previous studies indicated urban land use can increase sediment supply and pollutant transport when impervious surfaces and engineered water infrastructure increase the volume and velocity of surface runoff, stream power, and transport capacity [24]. Increased stream power associated with large storms can alter channel geomorphology via channel widening, channel incision, bank erosion and mass wasting [24].

For example, Gellis et al. [13] showed Tropical Storm Lee (a recent 100-year storm) caused stream bank erosion that accounted for 70% of suspended sediment concentration in the suburban and urban Upper Difficult Run, Virginia, USA. Results from Gellis et al. [13] emphasized the importance of streambank sources of sediment supply in suburban and urban watersheds. Thus, results from previous studies indicate streambank stabilization efforts are often necessary in urbanizing watersheds [13,14,24].

The combined influence of agricultural and urban land use practices on channel geomorphology and substrate composition is not well-understood. Investigations that quantitatively characterize land use influence on channel geomorphology and stream substrate composition in representative watersheds could help guide regional stream restoration efforts [4,26]. Therefore, the overall objective of this study was to quantify general trends and relationships between land use, stream hydromorphology, and substrate composition using observed data collected during a physical habitat assessment (PHA) in a representative experimental watershed located in the Midwestern USA. Sub-objectives were to (1) present observed spatial variability of channel geomorphology metrics, substrate frequency, and substrate embeddedness, and (2) quantify relationships between land use, channel geomorphology, and substrate frequency and embeddedness. Additionally, results are discussed with implications for watershed management in similar watersheds where agricultural and urban land uses have exacerbated problems with channel geomorphology alteration and streambed sedimentation.

## 2. Methods

### 2.1. Study Site

Hinkson Creek Watershed (HCW), is a rapidly urbanizing mixed-land use (forest, agriculture, urban) watershed located in the Lower Missouri Mississippi River Basin (LMMRB) (Table 1, Figure 1). Drainage area of HCW is approximately 230 km$^2$. Elevation ranges from 274 m above mean sea level (AMSL) in the headwaters to 177 m AMSL near the watershed outlet. At the time of this study, agricultural land use accounted for 36.7% HCW (28.4% grazing pasture and 8.3% row crop). Urban and suburban areas accounted for 29.0% of the total land use. Approximately 28 km$^2$ of impervious surfaces associated with urban land use were located within the municipal boundary of The City of Columbia (population 121,717; USCB [27]).

Soils in the upper elevations of HCW are generally loamy loess with a well-developed underling claypan in the argillic horizon of smectitic mineralization corresponding to the Mexico-Leonard association [28]. Claypan soils have increased surface runoff potential [29]. Soils that consist of a cherty clay solution residuum corresponding to the Weller-Bardley-Clinkenbeard (CBC) association are found in the lower reaches (upland of the alluvium). Soils in HCW are underlain by Burlington formation Mississippian series limestone in the lower reaches [28–30].

Climate in HCW is dominated by continental polar air masses in the winter and maritime and continental tropical air masses in the summer [31]. A 20-year climate record (2000–2019) obtained from the Sanborn Field climate station located on University of Missouri campus showed mean annual total precipitation was 962 mm and mean annual air temperature was 13.5 °C. A wet season occurs primarily during March through June [31,32]. The HCW main drainage, Hinkson Creek, is primarily stormflow dominated with a base flow index (ratio of base flow to total streamflow) of approximately 0.25 calculated using daily discharge data collected at a U.S. Geological Survey (USGS) gaging station (USGS 06910230, Site #4 in Figure 1) that has been intermediately monitoring stage since 1967.

During winter 2008, HCW was instrumented with a nested-scale experimental study design to investigate land use alterations to water quantity and quality [31]. Gauging sites (n = 5) partitioned the catchment into five sub-basins, each with different dominant land uses [31]. Site #5 was located near the watershed outlet, and Sites #1 to #4 were nested within (Figure 1). Site #1 (57.0% agricultural, 35.9% forested, 4.7% urban) was located in the agricultural land use dominated headwaters. Moving downstream from agricultural Site #1, agricultural land use decreased as urban land use increased. Site #3 (49.5% agricultural; 35.4% forested; 13.0% urban) was located at the rural-urban interface of HCW.

Site #5 (39.4% agricultural; 33.1% forested; 26.5% urban) is located in the lower most urbanized reaches. Results from previous studies showed land use alterations to the flow regime [28,33–35], environmental flows [36], stream temperature [37,38], suspended sediment [14,39,40], nutrients (i.e., total inorganic nitrogen species, total phosphorus) [41], and chloride [42] in HCW.

**Table 1.** Cumulative land use and land cover (LULC), drainage area, and stream length corresponding to each gauging site located in Hinkson Creek Watershed (HCW), Missouri, USA. Percent cumulative LULC is shown parenthetically.

| Variable | Site #1 | Site #2 | Site #3 [km² (%)] | Site #4 | Site #5 | HCW |
|---|---|---|---|---|---|---|
| Agricultural | 45.0 (57.0) | 56.4 (54.9) | 57.6 (49.5) | 78.5 (44.1) | 79.7 (39.4) | 85.4 (36.7) |
| Forested | 28.4 (35.9) | 37.5 (36.4) | 41.1 (35.4) | 62.8 (34.5) | 68.6 (33.1) | 74.9 (32.2) |
| Urban | 3.7 (4.7) | 6.6 (6.4) | 15 (13.0) | 37.1 (20.4) | 54.9 (26.5) | 67.6 (29.0) |
| Wetland | 1.9 (2.4) | 2.4 (2.3) | 2.5 (2.1) | 3.6 (2.0) | 4.4 (2.0) | 4.9 (2.1) |
| Total area | 79.0 | 102.9 | 116.2 | 182.0 | 207.5 | 232.8 |
| Stream length [§] | 22.8 | 29.8 | 35.4 | 43.6 | 53.0 | 56.1 |

[§] Stream length is shown in km.

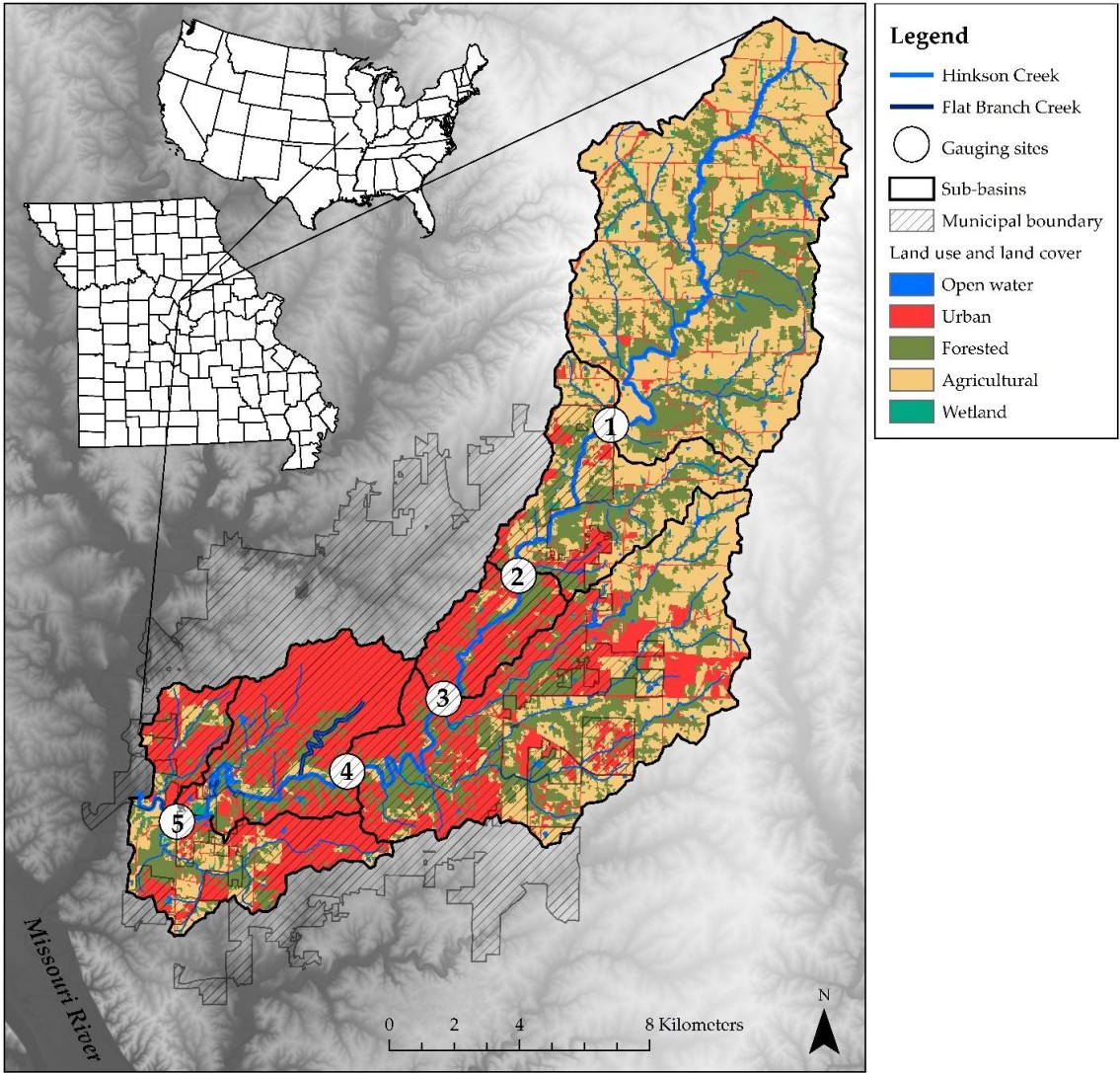

**Figure 1.** Land use and land cover of Hinkson Creek Watershed, Missouri, USA. Five nested gauging sites (numbered 1 to 5) and corresponding sub-basins are shown. Site #5 was located near the watershed outlet, Sites #1 to #4 nested within.

*2.2. PHA Data Collection and Analysis*

A physical habitat assessment (PHA) was performed in HCW during the study period (2013–2014) [43–45]. During a single survey, channel geomorphology and stream substrate data were collected at survey points (n = 561) every 100 m along the entire 56 km length of Hinkson Creek. At each survey point, data were collected (using a clinometer, extension pole, laser level, a laser range finder and/or meter stick) at principal transects that spanned from stream bank to bank perpendicular to flow. Additionally, data were collected at transects located 5 m upstream and downstream of the principal transect. Upstream and downstream transects were parallel to the principal transect. At any confluence of the main channel, three transects (upstream, downstream, and upstream in the tributary) were located equidistant from the center of the confluence.

### 2.2.1. Observed Channel Geomorphology

Channel width, wetted width of the stream, bankfull width, bank angle, bank height, and channel depth were measured using a laser level and/or laser range finder at each principal transect (n = 561). Following methods suggested by Harrelson et al. [46], physical indicators of bankfull level included the top of pointbars, changes in vegetation from aquatic to terrestrial, changes in slope, changes in bank material (e.g., from coarse gravel to sand), bank undercuts, or stain lines on bedrock or boulders. Bankfull width was measured parallel of the stream surface and perpendicular to stream flow from the lowest bank (i.e., bankfull bank) to the opposite bank.

Observed channel geomorphology data (collected every 100 m) dependent on drainage area were normalized as per methods used by Yanites and Tucker [47] using the following equation:

$$Ch^* = \frac{Ch_m}{kD_A^b} \tag{1}$$

where $Ch^*$ is area-normalized channel geomorphology, $Ch_m$ is the measured channel morphology metric, $D_A$ is drainage area, and $k$ and $b$ are fitted power regression coefficients.

Channel geomorphology data were reduced by averaging within five nested HCW sub-basins (Figure 1) to quantify the change in channel geomorphology with downstream distance across a rural-urban land use gradient. Sub-basins were delineated between nested gauging sites, and thus, were not cumulative watersheds. For example, channel width at Site #2 reflected the average of all the channel width measurements collected every 100 m upstream of Site #2 and downstream of Site #1. One-way Analysis of Variance (ANOVA) and Tukey Kramer post-hoc multiple comparison tests were used to test for significant differences (CI = 95%, p < 0.05) in average channel geomorphology metrics (i.e., channel width, bank height, bank angle, etc.) between five HCW sub-basins [48]. Tukey Kramer post-hoc multiple comparison test was selected to elucidate site differences with a narrow confidence interval [49].

### 2.2.2. Observed Streambed Substrate Frequency and Embeddedness

Substrate particle size-class (Table 2) was estimated at each principal, upstream, downstream transect following methods suggested by Peck et al. [50], and Wolman [51]. At each transect, particles were sampled in five locations from bank to bank (left bank, left center, center, right center, and right bank) perpendicular to flow for a total of 8415 stream substrate samples. Substrate frequency was quantified as a percent of each individual substrate type relative to the total number of substrate samples collected within the area of interest. Additionally, substrate embeddedness of each particle was estimated as the percent vertical entrainment of a streambed substrate sample. Channel unit type (i.e., trench pool, plunge pool, impoundment pool, pool, split channel, riffle, glide, dry channel, etc.) was also recorded at each principal transect as per methods used by Peck et al. [50]. While the influence of water depth on channel unit identification was not quantified in the current study, adverse effects were assumed negligible considering the water depth recorded in the thalweg ranged from 0 to

280 cm with a median of 40 cm at the time of channel unit identification. Channel units were grouped into general categories of pool, riffle, and glide for further analysis. Observed substrate particle size frequency and percent embeddedness data were reduced by averaging in channel units (i.e., pools, riffles, glides, and dry channels), from bank to bank, and within each HCW sub-basin to quantify the change in stream substrate composition associated with riffle-pool habitat with downstream distance across a rural-urban land use gradient. Sub-basins were delineated between nested gauging sites and were thus not cumulative.

**Table 2.** Definition of streambed substrate variables used in this study.

| Code | Description |
| --- | --- |
| LWD PILE | Large woody debris |
| VG | Vegetation |
| SH | Shell |
| CU | Culvert |
| BR | Bridge |
| RP | Rip-rap |
| WD | Wood |
| RS | Smooth Bedrock (>4000 mm) |
| RR | Rough Bedrock (>4000 mm) |
| RC | Concrete/asphalt (>4000 mm) |
| XB | Large boulder (1000 to 4000 mm) |
| SB | Small boulder (256 to 1000 mm) |
| CB | Cobble (64 to 256 mm) |
| GC | Coarse gravel (16 to 64 mm) |
| GF | Fine gravel (2 to 16 mm) |
| SA | Sand (0.06 to 2 mm) |
| FN | Silt/clay/muck (<0.06 mm) |

## 3. Results

### 3.1. Observed Geomorphology

Results from power regression analysis showed drainage area explained 41.2% of the variance in channel width, 58.6% of the variance in bankfull width, 29.9% of the variance in bank height, and 42.1% of the variance in bankfull depth ($p < 0.001$; n = 561) (Figure 2). After channel width and height and bankfull width and depth metrics were area-normalized, results showed significant differences in channel morphology metrics averaged within five sub-basins of HCW (Table 3). For example, area-normalized average channel width ranged from 0.12 to 0.19 m km$^{-2}$. Area-normalized average bankfull width ranged from 0.21 to 0.27 m km$^{-2}$. Area-normalized average bank height ranged from 0.38 to 0.56 m km$^{-2}$. Area-normalized average bankfull depth ranged from 0.34 to 0.49 m km$^{-2}$. Average minimum and maximum bank angle ranged from 28.0 to 47.8 degrees. Average width to depth ratios and bed slope ranged from 6.3 to 10.3, and 0.14 to 0.22%, respectively. These results quantitatively characterized channel geomorphology metrics important for stream restoration efforts in HCW (Table 3).

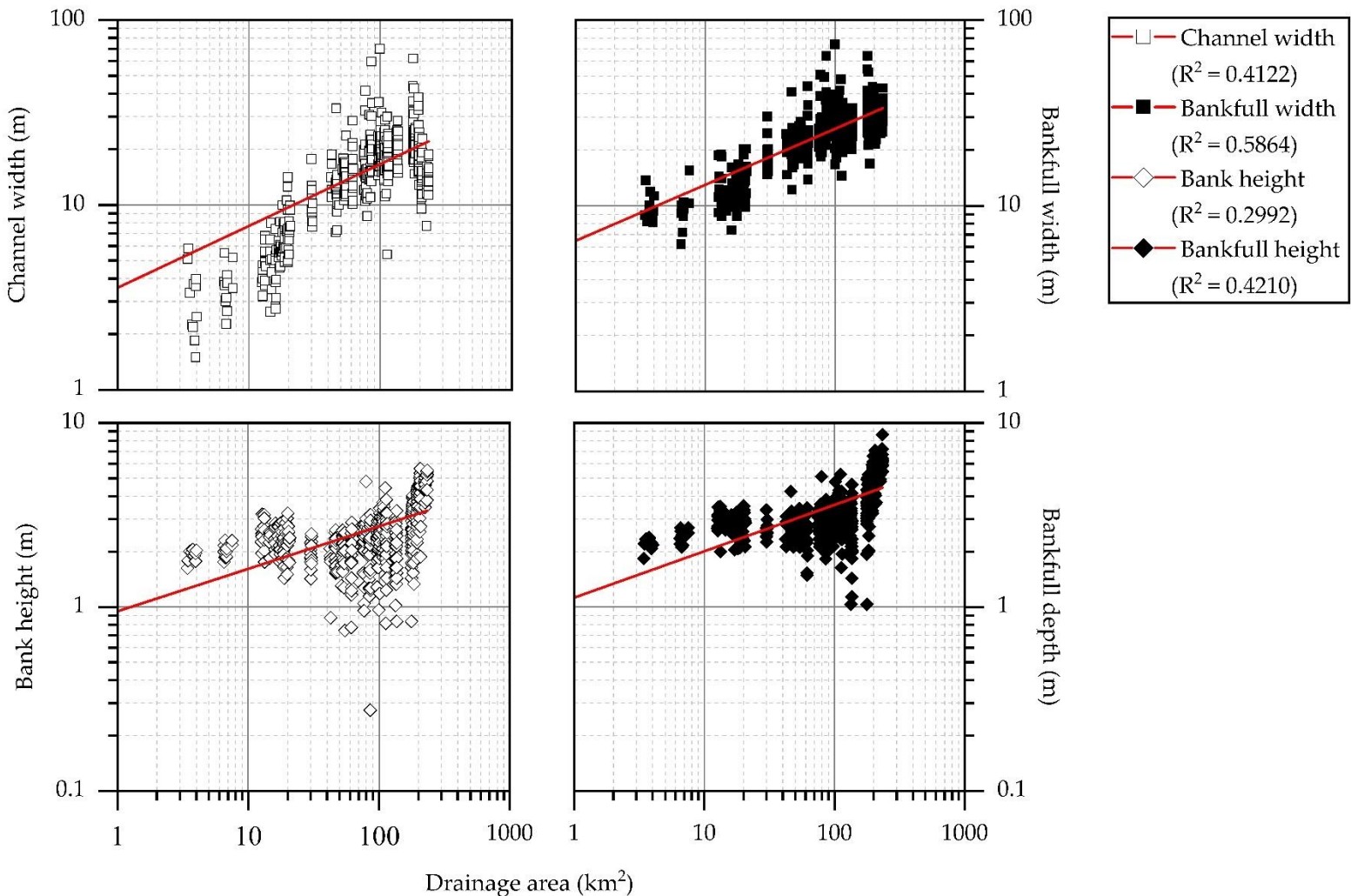

**Figure 2.** Power relationships between drainage area and observed channel morphological metrics in Hinkson Creek Watershed, Missouri, USA.

**Table 3.** Channel morphology metrics measured at survey sites (n = 561) and averaged within five nested sub-basins located in Hinkson Creek Watershed, Missouri, USA. Sub-basins were delineated between nested gauging sites and were thus not cumulative.

| Variable [§] | Site #1 | Site #2 | Site #3 | Site #4 | Site #5 |
|---|---|---|---|---|---|
| Channel width (m km$^{-2}$) [†] | 0.14 [abc] | 0.19 [ade] | 0.15 [df] | 0.16 [bg] | 0.12 [cefg] |
| Bankfull width (m km$^{-2}$) [†] | 0.23 [a] | 0.27 [abcd] | 0.23 [b] | 0.23 [c] | 0.21 [d] |
| Bank height (m km$^{-2}$) [†] | 0.47 [abc] | 0.39 [ad] | 0.44 [e] | 0.38 [bf] | 0.56 [cdef] |
| Bankfull depth (m km$^{-2}$) [†] | 0.42 [abcd] | 0.36 [ae] | 0.37 [bf] | 0.34 [cg] | 0.49 [defg] |
| Min bank angle (°) | 28.0 [ab] | 22.5 [ac] | 22.0 [bd] | 25.4 | 28.0 [cd] |
| Max bank angle (°) | 47.8 [abc] | 38.7 [a] | 40.3 [b] | 39.9 [c] | 44.4 |
| Width: depth (–) | 6.3 [abc] | 10.2 [ade] | 8.5 [bdfg] | 10.3 [cfh] | 6.6 [egh] |
| Bed slope (%) | 0.14 [abcd] | 0.22 [aef] | 0.24 [agh] | 0.18 [aeg] | 0.16 [dfh] |

[§] Averages with corresponding letters ([a–h]) indicate significant differences (α = 0.05).  [†] Channel geomorphic variables normalized by drainage area.

No significant relationships were observed between any one land use and land cover (LULC) index shown in Table 1 and average channel morphology metrics at five sub-basins in HCW ($R^2 \leq 0.593$; $p \geq 0.080$; n = 5). However, multiple linear regression (MLR) results showed agricultural and urban land use combined explained nearly all of the variance in average width to depth ratios ($R^2 = 0.960$; $p = 0.020$; n = 5), and maximum bank angle ($R^2 = 0.896$; $p = 0.052$; n = 5). Thus, results implied the expected relationships between drainage area and channel morphology were altered, at least in part, by the combined influence of agricultural and urban land use in HCW.

*3.2. Observed Streambed Substrate Frequency and Embeddedness*

Results from stream substrate frequency analysis quantitatively characterized bank to bank variability in streambed substrate composition important for stream physical habitat and aquatic refugia in HCW. Results showed percent fine substrate (FN) summarized for each bank to bank position along the entire drainage ranged from 11.6% at the center channel to 62.0% on the right descending bank. Sand substrate (SA) ranged from 17.3% on the left bank to 33.9% center channel. Percent fine gravel (GF), course gravel (GC), cobble (CB), and small boulder (SB) followed a similar trend as sand substrate from bank to bank. For example, GC ranged from 3.3% at the right bank to 21.3% at the center channel position. There was no obvious bank to bank trend in substrate frequency large boulders (XB), rough bedrock (RR), smooth bedrock (RS), or the rest of the substrate types presented in Table 4.

Examination of substrate frequency at five sub-basins showed the frequency of substrate types smaller than GC decreased with downstream distance from agricultural Site #1 in the headwaters to urban Site #3 at the rural-urban interface of the watershed. Continuing downstream, substrate frequency of substrate types smaller than GC increased from Site #3 to urban Site #5 located near the watershed outlet. For example, FN, SA, and GF decreased from Site #1 to Site #3 by 17%, 19%, and 3%, respectively. Continuing downstream, FN, SA, and GF increased from Site #3 to Site #5 by 19%, 12%, and 3%, respectively. Conversely, some of the larger substrate types (i.e., GC, CB, SB, and XB) increased with downstream distance from Site #1 to Site #3, and then, decreased from Site #3 to Site #5. Observed substrate composition data showed increased bed rock channel constraints at Sites #2 to #4 (9 to 11% RR and RS) located in the mid-reaches relative to Site #1 (3% RR) located in the agricultural headwaters and Site #5 (1% RR and RS) in the lower urban reaches (Table 5). These results quantifiably characterized the change of stream substrate frequency across an agricultural-urban land use gradient, and in synthesis, indicated increased substrate embeddedness in the agricultural headwaters and lower urban reaches located outside of the observed bedrock channel constraints in the mid-reaches of the study catchment.

**Table 4.** Bank to bank variability of streambed substrate frequency in Hinkson Creek Watershed, Missouri, USA. Each datum is a percent of a substrate type observed at a bank position along the entire 56km stream length of the reference stream. Substrate definitions are presented in Table 2.

| Substrate Type | Left Bank | Left Center | Center (%) | Right Center | Right Bank |
|---|---|---|---|---|---|
| Silt/clay/muck (FN) | 57.9 | 17.5 | 11.6 | 17.6 | 62.0 |
| Sand (SA) | 17.3 | 28.8 | 33.9 | 34.3 | 17.5 |
| Fine gravel (GF) | 0.2 | 5.1 | 6.1 | 4.1 | 0.4 |
| Coarse gravel (GC) | 3.5 | 20.1 | 21.3 | 16.8 | 3.3 |
| Cobble (CB) | 5.6 | 13.0 | 13.8 | 15.9 | 4.1 |
| Small boulder (SB) | 3.7 | 4.3 | 4.3 | 4.3 | 3.0 |
| Large boulder (XB) | 1.4 | 0.6 | 1.2 | 0.9 | 1.2 |
| Rough Bedrock (RR) | 4.2 | 5.9 | 3.7 | 2.5 | 1.2 |
| Smooth Bedrock (RS) | 1.5 | 2.0 | 1.4 | 1.1 | 1.1 |
| Wood (WD) | 2.9 | 0.4 | 0.6 | 0.4 | 2.5 |
| Rip-rap (RP) | 0.8 | 0.3 | 0.1 | 0.1 | 1.7 |
| Bridge (BR) | – | – | 0.1 | – | – |
| Culvert (CU) | – | – | – | – | – |
| Concrete/asphalt (RC) | 0.1 | – | – | – | 0.3 |
| Shell (SH) | – | – | 0.1 | – | – |
| Vegetation (VG) | 0.9 | 2.0 | 1.6 | 2.1 | 1.8 |
| Large woody debris (LWD PILE) | 0.1 | 0.1 | 0.1 | 0.1 | – |

**Table 5.** Streambed substrate frequency observed within five nested sub-basins in Hinkson Creek Watershed, Missouri, USA. Each datum is a percent of a substrate type. Sub-basins were delineated between nested gauging sites, and thus, were not cumulative watersheds excepting Site #1 located in the headwaters. Substrate definitions are presented in Table 2.

| Substrate Type | Site #1 | Site #2 | Site #3 (%) | Site #4 | Site #5 |
|---|---|---|---|---|---|
| Silt/clay/muck (FN) | 40 | 26 | 23 | 9 | 42 |
| Sand (SA) | 32 | 18 | 13 | 29 | 25 |
| Fine gravel (GF) | 4 | 1 | 1 | 3 | 4 |
| Coarse gravel (GC) | 9 | 19 | 16 | 18 | 14 |
| Cobble (CB) | 7 | 16 | 21 | 17 | 5 |
| Small boulder (SB) | 2 | 5 | 9 | 10 | 2 |
| Large boulder (XB) | 0 | 1 | 3 | 3 | 1 |
| Rough Bedrock (RR) | 3 | 10 | 9 | 1 | 1 |
| Smooth Bedrock (RS) | 0 | 0 | 2 | 8 | 0 |
| Wood (WD) | 2 | 2 | 0 | 0 | 1 |
| Rip-rap (RP) | 0 | 1 | 1 | 0 | 1 |
| Bridge (BR) | – | – | 0 | – | – |
| Culvert (CU) | – | – | – | – | – |
| Concrete/asphalt (RC) | 0 | – | – | – | 0 |
| Shell (SH) | – | – | 0 | – | – |
| Vegetation (VG) | 2 | 2 | 1 | 0 | 4 |
| Large woody debris (LWD PILE) | 0 | 0 | 0 | 0 | – |

Substrate embeddedness also varied between different stream habitats (i.e., riffle, glide, and pool). For example, results showed that pools (74.9% embeddedness) were associated with significantly greater substrate embeddedness than riffles (58.5% embeddedness) and glides (60.1% embeddedness) ($p = 0.0014$; CI = 95%). However, no significant differences were observed between riffles and glides ($p = 0.9367$; CI = 95%). Significant differences in substrate embeddedness were especially apparent at Sites #1 and #5 compared to the mid-reaches (i.e., Sites #2 to #4) ($p = 0.013$; CI = 95%) (Table 6). Stream banks were embedded with FN and SA in more than 88% of pools habitat at Sites #1 and #5. Even greater were differences in substrate embeddedness at center of channel samples (i.e., left center, center, and right center) of pools at Sites #1 and #5 (70 to 79%) compared to Sites #2 to #4 (39 to 57%). Center of channel samples of riffles ranged from 51 to 72% at Sites #1 and #5, and 27 to 46% at Sites #2 to #4.

Due to the observed widespread substrate embeddedness, results generally showed reduced spatial complexity of substrate composition in riffles, glides, and pools of HCW.

**Table 6.** Bank to bank variability of streambed substrate embeddedness in riffles, glides, and pools measured at survey sites (n = 561) and averaged within five nested sub-basins located in Hinkson Creek Watershed, Missouri, USA. Each datum shows substrate embeddedness as a percent. Sub-basins were delineated between nested gauging sites, and thus, were not cumulative watersheds excepting Site #1 located in the headwaters.

| Bank Position | Site (#) | Riffles | Glides (%) * | Pools |
|---|---|---|---|---|
| Left bank | Site #1 | 83 | 100 | 88 |
| | Site #2 | 76 | 67 | 71 |
| | Site #3 | 76 | 81 | 67 |
| | Site #4 | 72 | 71 | 72 |
| | Site #5 | 85 | 65 | 88 |
| Left center | Site #1 | 56 | 91 | 75 |
| | Site #2 | 27 | 23 | 42 |
| | Site #3 | 39 | 56 | 44 |
| | Site #4 | 46 | 59 | 54 |
| | Site #5 | 59 | 32 | 70 |
| Center | Site #1 | 51 | 80 | 76 |
| | Site #2 | 31 | 18 | 46 |
| | Site #3 | 40 | 39 | 39 |
| | Site #4 | 43 | 51 | 52 |
| | Site #5 | 61 | 70 | 75 |
| Right center | Site #1 | 52 | 93 | 79 |
| | Site #2 | 30 | 27 | 57 |
| | Site #3 | 34 | 43 | 47 |
| | Site #4 | 41 | 59 | 52 |
| | Site #5 | 72 | 42 | 76 |
| Right bank | Site #1 | 70 | 100 | 90 |
| | Site #2 | 77 | 78 | 80 |
| | Site #3 | 70 | 69 | 86 |
| | Site #4 | 65 | 73 | 77 |
| | Site #5 | 88 | 78 | 90 |

* Percent embeddedness of glides at Site #1 indicate dry channel units.

Substrate embeddedness was significantly correlated at the 0.05 level with 4 of 8 channel morphology metrics considered in this study. For example, minimum bank angle ($R^2 = 0.8511$; $p = 0.0164$; n = 5), maximum bank angle ($R^2 = 0.8400$; $p = 0.0183$; n = 5), width to depth ratio ($R^2 = 0.7225$; $p = 0.0424$; n = 5), and bed slope ($R^2 = 0.7424$; $p = 0.0384$; n = 5) explained a substantial amount of variance in substrate embeddedness between sites in HCW. Minimum and maximum bank angle were positive correlates. Width to depth ratios and bed slope were negative correlates. Relationships between drainage area and land use were not significant at a 0.05 level. However, there was a general trend for increased substrate embeddedness in the agricultural headwaters at Site #1 and in the lower urban reaches at Site #5 (80% embedded) relative to the mid-reaches at Sites #2 to #4 (56 to 60% embedded) of HCW.

There was also a general trend for increased frequency of pool habitat coupled to decreased frequency of riffle and glide habitats in the agricultural headwaters and in the lower urban reaches of HCW (Table 7). The frequency of pools ranged from 56.1% at the urban Site #4 of HCW to 85.7% at urban Site #5. Conversely, the frequency of riffle habitat increased from 12.2% at Site #5 in the lower urban reaches to 28.0% at urban Site #4. The frequencies of pool, riffle, and glide habitats were not significantly correlated with substrate embeddedness ($R^2 \geq 0.420$; $p \leq 0.140$; n = 5). However, results showed strong correlations between channel unit frequency and frequency of fine substrate type ($0.927 \leq R^2 \geq 0.854$; $p \leq 0.016$; n = 5) highlighting observed streambed siltation in Hinkson Creek.

**Table 7.** Channel unit frequency measured at survey sites (n = 561) and averaged within five nested sub-basins located in Hinkson Creek Watershed, Missouri, USA.

| Site (#) | Pools | Riffles (%) | Glides | Dry |
|----------|-------|-------------|--------|-----|
| Site #1 | 78.6 | 13.1 | 0.0 | 8.3 |
| Site #2 | 72.1 | 23.6 | 4.3 | 0.0 |
| Site #3 | 64.3 | 25.0 | 10.7 | 0.0 |
| Site #4 | 56.1 | 28.0 | 15.9 | 0.0 |
| Site #5 | 85.7 | 12.2 | 2.1 | 0.0 |

## 4. Discussion

### 4.1. Observed Geomorphology

Results from this study show how expected longitudinal change in channel geomorphology can be altered independently or by combined bedrock channel constraints and human activity. For example, the mid-reaches of HCW were associated with increased bedrock channel constraints (Figure 3), which according to Montgomery et al. [52] indicates transport capacity in excess of sediment supply. Outside of the mid-reaches (i.e., Sites #1 and #5) where less than 3% presence of channel bedrock constraints were observed, signs of channel incision were apparent. Site #1 in the agricultural headwaters and Site #5 located the lower urban reaches of HCW (Figure 3). Agricultural and urban land use explained nearly all the variance in average width to depth ratios, and maximum bank angle ($0.896 \leq R^2 \geq 0.960$; $0.020 \leq p \geq 0.052$; n = 5). At Site #1 located in the agricultural headwaters (drainage area = 79.0 km$^2$), width to depth ratios increased at a rate of about 0.3 km$^{-1}$, and maximum bank angle decreased by about 0.5 degrees km$^{-1}$ as agricultural land use decreased from 100 to 56.9%, forested land use increased from 0 to 35.9%, and urban land use increased from 0 to 4.7% over 22.8 km stream distance (Figure 4). With stream distance from agricultural Site #1 to sub-urban Site #3 located at the rural-urban interface of HCW, (drainage area = 116.2.0 km$^2$), width to depth ratios increased by a rate of about 0.01 m m$^{-1}$ km$^{-1}$, and maximum bank angle decreased by about 0.1 degrees km$^{-1}$ as agricultural land use decreased to 49.5%, forested land use decreased by 0.5%, and urban land use increased to 13.0% over 12.6 km stream distance (Figure 4). Continuing downstream from sub-urban Site #3 to urban Site #5 located in the lower urban reaches (drainage area = 207.5 km$^2$), width to depth ratios decreased by about 0.2 km$^{-1}$, and maximum bank angle increased by about 0.2 degrees km$^{-1}$ as agricultural land use decreased to 38.4%, forested land use decreased to 33.1%, and urban land use increased to 26.5% over 17.6 km stream distance (Figure 4). Bankfull depth increased by 0.13 m km$^{-1}$ at Site #1, 0.02 m km$^{-1}$ between Sites #1 and #3, and 0.15 m km$^{-1}$ between Sites #3 and #5. In combination, these results indicated increased channel incision at Site #1 in the agricultural headwaters where agricultural land use accounted for greater than 50% of total catchment area, and Site #5 the lower urban reaches where urban land use accounted for greater than 20% of total catchment area in HCW. Thus, Sites #1 and #5 were considered hot spots of channel incision due to increased rates of change in channel geomorphology metrics.

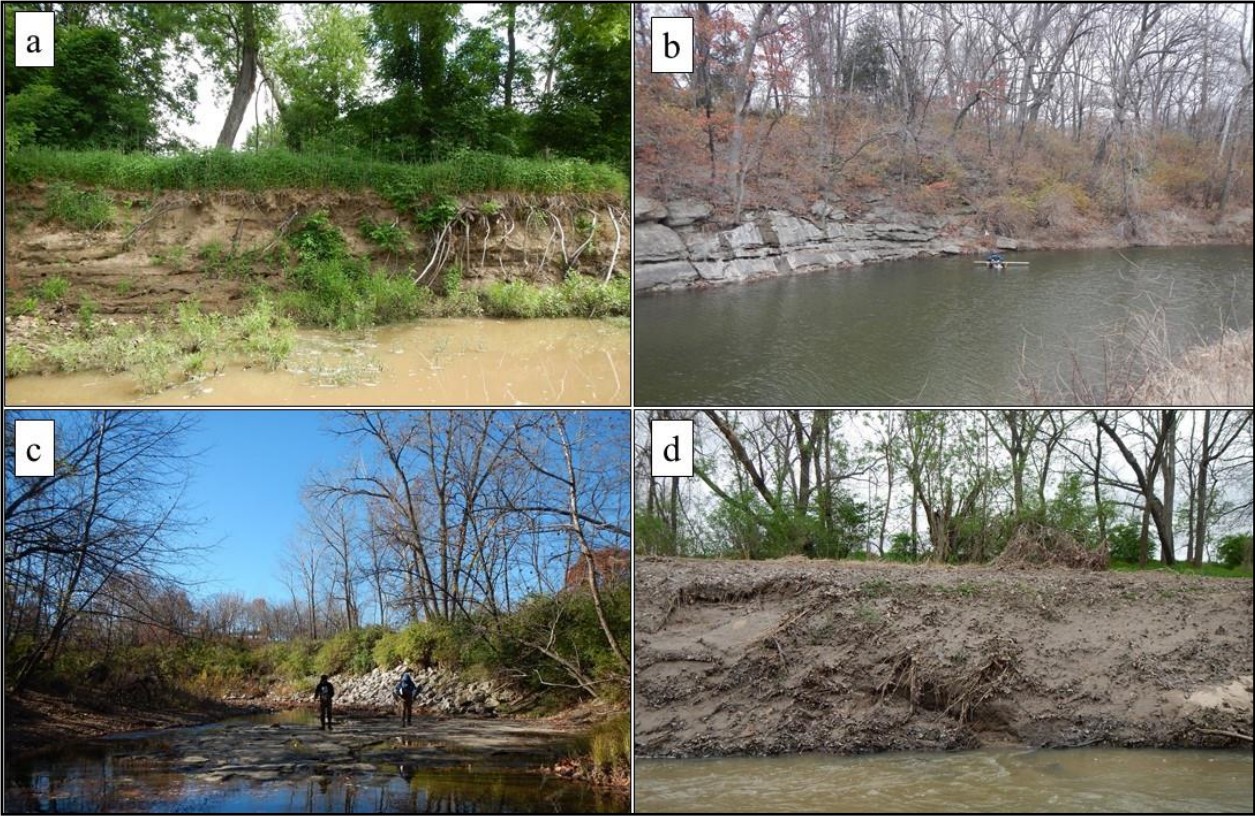

**Figure 3.** Examples of channel characteristics associated with the agricultural headwaters (**a**), rural-urban mid-reaches (**b,c**), and lower urban reaches (**d**) of Hinkson Creek Watershed, USA.

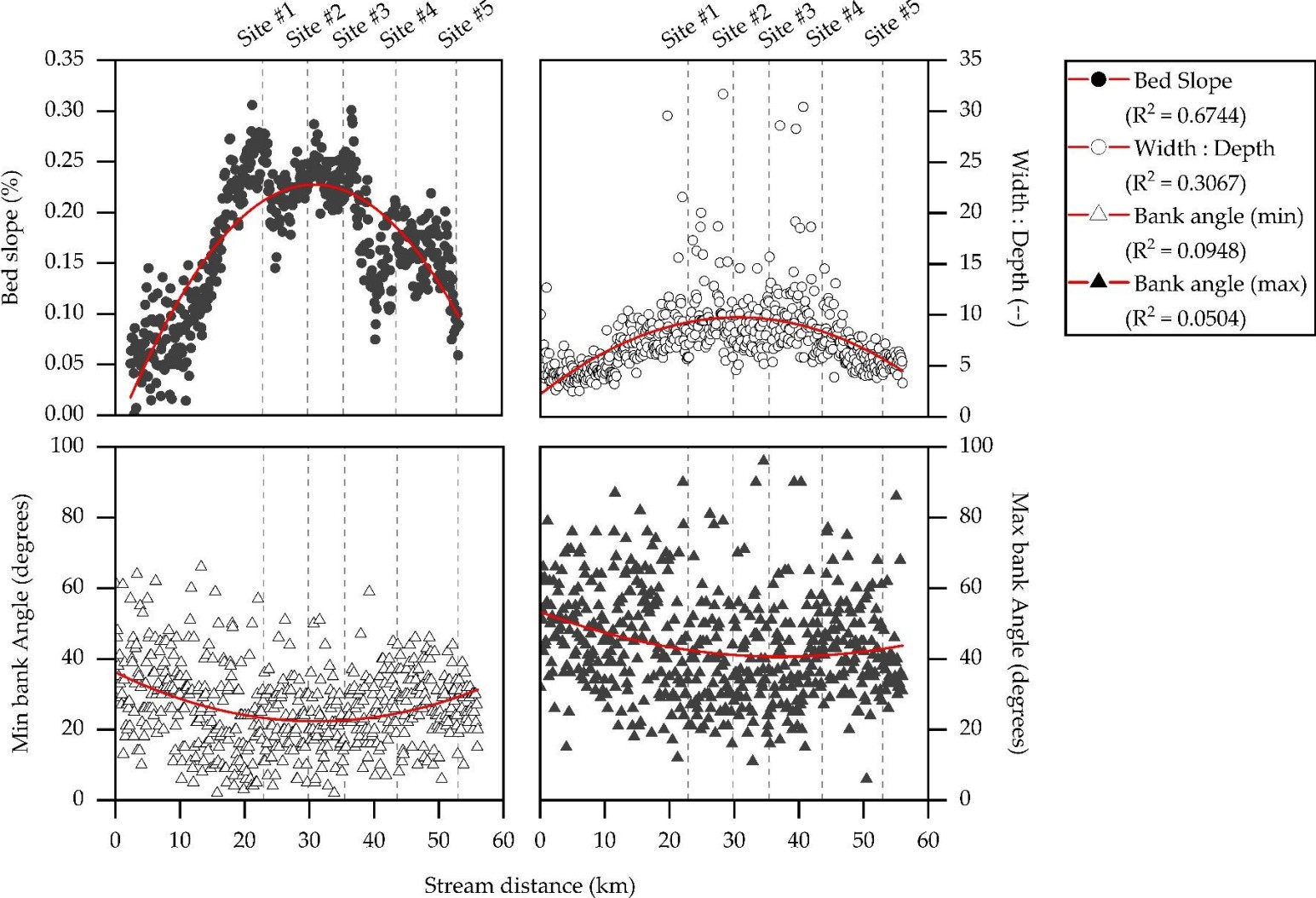

**Figure 4.** Trend lines associated with channel morphology variables with downstream distance (km) in Hinkson Creek Watershed, Missouri, USA. Vertical lines show location of five nested gauging sites.

Previous studies have shown agricultural land use can result in channel incision through various mechanisms including, but not limited to, channelization, deforestation, and alterations to soil and channel hydraulics [53–56]. Simon et al. [56] noted increased pore water pressure coupled to decreased sheer strength can lead to streambank erosion. Given the influence of soil hydraulic forces on streambank stability, agricultural areas are subject to increased bank failure where crop irrigation lowers water table levels in the vicinity of a stream. A study by Zaimes et al. [54] showed incised stream channels were associated with increased bank mass wasting, streambank erosion and sediment load in Beak Creek, an agricultural land use dominated Midwestern stream located in central Iowa, USA. Streambank erosion rates varied from meandering row crop fields (387 mm year$^{-1}$), cattle and horse pastures (295 mm year$^{-1}$), and meandering forest buffered reaches (142 mm year$^{-1}$). Results indicated that forested riparian buffers would reduce streambank erosion by 72%. Streambank stabilizing root systems associated with riparian vegetation add roughness, reduce stream power, and enhance bank accretion [56]. Results from the aforementioned studies are in agreement with results from the current work showing agricultural land use alterations to channel morphology of Midwestern streams.

Previous studies have also shown urban land use can cause channel incision via alterations to streamflow regimes [24,25,57–59]. A literature review by Walsh et al. [24] showed increased impervious surfaces associated with urban land use can cause increased volume and velocity of surface runoff and a flashy hydrologic streamflow response linked to increased bank wasting, channel incision, and scouring. For example, Jordan et al. [60] showed urban land use alterations to flow caused a 9 to 61% increase of sediment yield due to channel incision and bank erosion in Berryessa Creek, California, USA. A study by Shields et al. [59] quantified differences in channel incision, streamflow, water quality, and stream physical habitat between rural and urban catchments located in the Yazoo River basin, Mississippi, USA. Results showed urban land use was associated with decreased physical aquatic habitat, 6.4 times median rate of rise, 1.8 times channel depth, 3.5 times channel width, 2 to 3 times turbidity and suspended solids, 2 times fish species, and 4 times the amount of fish biomass per unit of effort [59]. Results from the current work are a novel addition to previous studies considering the intensive sampling regimen (n = 561) that made possible the estimation of the rate of change in channel morphology and substrate composition across an agricultural-urban land use gradient.

*4.2. Observed Streambed Substrate Frequency and Embeddedness*

No channel morphology variable significantly explained the variance in substrate embeddedness of pools, riffles, and glides for each bank to bank sampling position (Figure 5) highlighting the spatial complexity of streambed substrate composition in this study. Spatial complexity of substrate composition was, in part, shaped by the thalweg which meandered from bank to bank. The thalweg was generally associated with decreased substrate embeddedness due to increased stream velocity, and thus, increased sediment transport capacity. While much of the observed bank to bank variability in substrate embeddedness was attributed to thalweg position, results from the current work in combination with previous research in the region indicated longitudinal variability in substrate composition was attributed to the presence claypan soils and agricultural land use in the headwaters, increased bed slope and bedrock channel constraints in the mid-reaches, and the influence of urban land use associated with increased impervious surfaces in the lower reaches (Figure 3).

Previous studies in HCW and elsewhere have shown claypan soils and agricultural land use are associated with increased surface runoff, soil erosion, and channel sediment supply [14,15,29]. Lerch et al. [29] noted claypan soils corresponding to the Mexico-Leonard association consisting of an argillic soil horizon of smectitic mineralogy with clay content of 450–650 g kg$^{-1}$ formed at 10 to 50 cm depth are characterized by increased surface runoff. Willett et al. [15] showed claypan soils were associated with increased bank sediment supply that accounted for 88% of total sediment supply. Streambank erosion was particularly high during winter months attributed to a combination of increased frequency and large magnitude flow events, freeze/thaw cycles, high antecedent moisture conditions, and lack of vegetation. In the current work, increased bank angle and substrate embeddedness were

apparent during field sampling in the agricultural headwaters of HCW where claypan soils are present. Results from the current work are among the first to quantitatively characterize agricultural land use influence on substrate frequency and substrate embeddedness in the Central Claypan Region and point to a need to mitigate the influence claypan soils and agricultural land use on degradation of stream hydrogeomorphology in HCW. Thus, results from this study in combination with previous research have implications in other agricultural watersheds where near surface soil features (e.g., claypans, argillic horizons, or fragipans) have increased sediment supply.

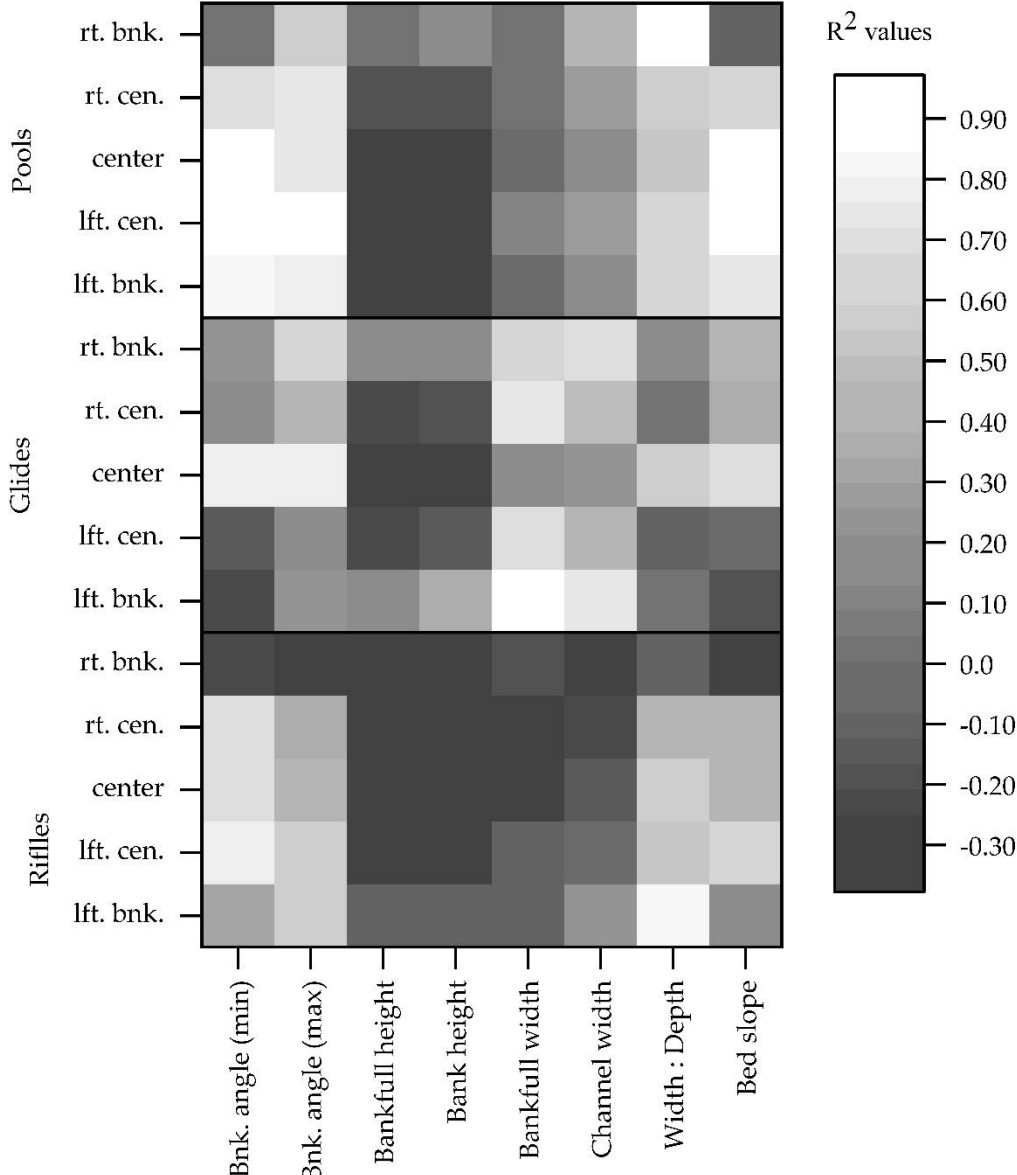

**Figure 5.** Explained variance between channel morphology metrics (x-axis) and substrate embeddedness from left bank (lft. bnk.) to right bank (rt. bnk.) associated with riffles, glides, and pools (y-axis) at five sub-basins of Hinkson Creek Watershed, Missouri, USA. No glide channel units were observed at Site #1, instead, percent embeddedness of dry channel units is shown.

While substrate embeddedness was greater in the agricultural headwaters, substrate embeddedness decreased as bed slope increased in the mid-reaches. Agricultural Site #1 was associated with 26 to 39% greater frequency of substrate smaller than GC, and 20 to 25% greater substrate embeddedness compared to Sites #2 to #4 in the mid-reaches of HCW (Figure 6). Bed slope, width to depth ratio, and the frequency

of substrate greater than GC were inversely related to substrate embeddedness (Figure 4). Process-based understanding of the control of channel morphology on streamflow and sediment transport [52] indicated the aforementioned general trends in the observed data were physically meaningful. Given that bed slope controls velocity of streamflow, stream capacity, and stream competence, it makes sense that Sites #2 to #4 located in the mid-reaches were generally associated with less substrate embeddedness and frequency of substrate less than GC in diameter compared to Sites #1 and #5. The mid-reaches of HCW were also associated with increased bedrock substrate and channel constraints (Figure 3), which indicated transport capacity in excess of sediment supply [52]. However, outside of the mid-reaches, increased substrate embeddedness was observed at agricultural Site # 1 and urban Site #5 (Figure 3).

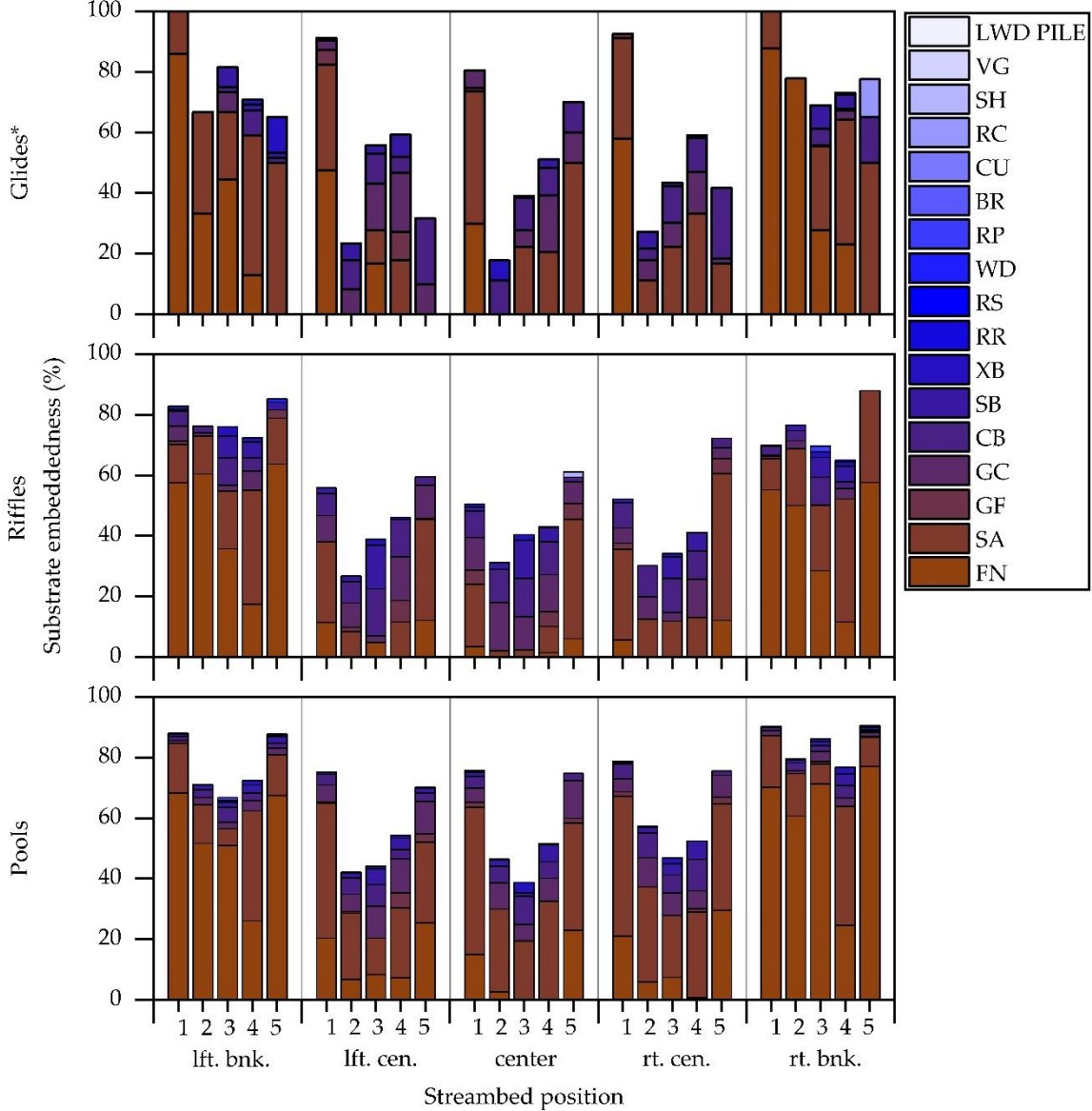

**Figure 6.** Substrate frequency and embeddedness from left bank (lft. bnk.) to right bank (rt. bnk.) associated with riffles, glides, and pools located within five nested sub-basins (numbered 1 to 5) of Hinkson Creek Watershed, Missouri, USA. Sub-basin #1 was located in the headwaters. Sub-basin #5 was located near the watershed outlet. No glide channel units were observed at Site #1, instead, percent embeddedness of dry channel units is shown. Substrate definitions are presented in Table 2.

The greatest rate of change in substrate embeddedness with downstream distance was observed in riffles located in the lower urban reaches where bank height exceeded 7 m (Figure 7). For example,

percent embeddedness of riffle habitat decreased by at a rate of about 2% km$^{-1}$ as agricultural land use decreased from 100 to 56.9%, forested land use increased from 0 to 35.9%, and urban land use increased from 0 to 4.7% over 22.8 km stream distance at Site #1 (Figure 7). However, percent embeddedness of riffles remained relatively constant (i.e., negligible rate of change) from agricultural Site #1 to sub-urban Site #3 located at the rural-urban interface where increased bedrock channel constraints were observed in HCW (Figure 7). Continuing downstream from sub-urban Site #3 to urban Site #4, substrate embeddedness of riffles began to increase by about 1.4% km$^{-1}$ over approximately 8 km of stream distance. Further downstream, percent embeddedness of riffles increased rapidly by 5.3% km$^{-1}$ between urban Sites #4 and #5 (Figure 7). Similar trends were observed in glides and pools as well (Figure 7). Thus, these results showed increased rate of change in percent embeddedness linked to degraded physical habitat (riffles, glides and pools) at agricultural Site #1 and urban Site #5, with a disproportionate rate of increase of substrate embeddedness in riffle habitat of the lower urban reaches pointing to symptoms of urban stream syndrome in HCW.

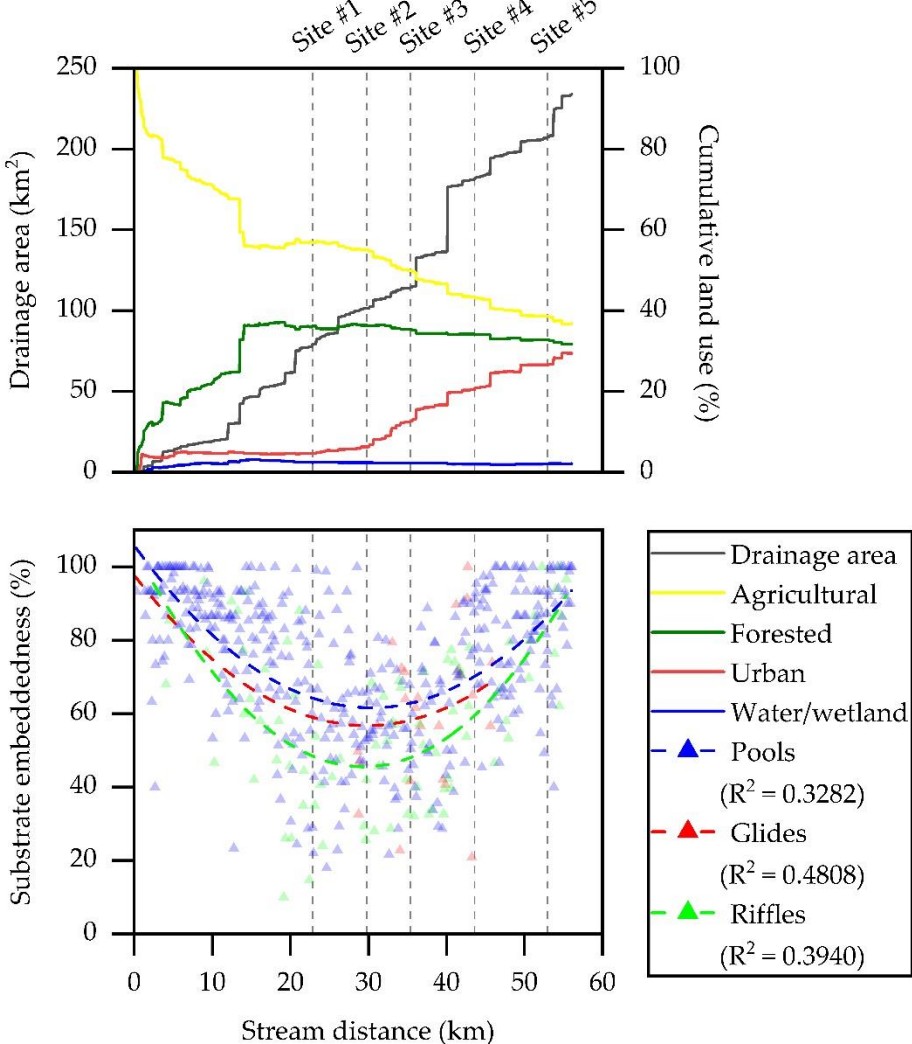

**Figure 7.** Watershed characteristics and trend lines associated with substrate embeddedness in pools, riffles, and glides with downstream distance (km) in Hinkson Creek Watershed, Missouri, USA. Vertical lines show location of five nested gauging sites.

The observed channel incision in the lower reaches of the current work was corroborated by other authors who also reported urban land use influence on channel morphology [24,60,61]. Blann et al. [53] discussed how increased channel incision disrupts hydrologically and ecologically

important stream-floodplain-riparian connectivity. In urban catchments, channel incision is often attributed to increased impervious surfaces and engineered waterways that connect impervious surfaces directly to stream channels [24,25]. Increased impervious surfaces have been shown to increase the volume and velocity of surface runoff in HCW [36] and elsewhere [24,25,62]. Increased surface runoff commonly translates to alterations to flow regimes (i.e., flow frequency, magnitude, timing, duration and rate of change), sediment transport regimes, water quality, and aquatic ecosystem health [24,25]. Clearly, there is a need to mitigate land use alterations to channel morphology via channel stabilization efforts in HCW and similar catchments globally. However, Vietz et al. [58] noted channel stabilization methods may not be sufficient to combat channel degradation in urbanized catchments. There is also a critical need for flow regime management efforts to reduce flow magnitude, frequency, and rate of change associated with alterations to channel morphology.

It was evident in the current work that simultaneously occurring agricultural and urban land uses exacerbated problems with substrate embeddedness in HCW particularly in the headwaters and lower urban reaches were bedrock constraints were less abundant. Previous studies also showed a general trend for suspended sediment and nutrients to decrease from Site #1 in the agricultural headwaters to Site #3 at the rural-urban interface of the watershed, and then, increase from Site #3 to urban Site #5 located near the watershed outlet in HCW [14,40,41]. In fact, significant relationships were observed between substrate embeddedness values reported in the current work and concentrations of suspended sediment ($R^2$ = 0.798; p = 0.026), nitrite-N ($R^2$ = 0.975; p = 0.001), and ammonia-N ($R^2$ = 0.956; p = 0.002) reported in previous studies in HCW [14,41]. Notably, suspended sediment and total phosphorous yields were particularly high compared to other regions within the Mississippi River Basin. For example, observed suspended sediment yields exceeding 300 Mg km$^{-2}$ year$^{-1}$ were 54%, 80%, and 87% greater than sediment yields from the Ohio River, the Upper Mississippi River, and the Missouri River basins, respectively [14]. Total phosphorous yields (0.979 kg ha$^{-1}$ year$^{-1}$) were also high for the region [41]. In combination with results of LULC alterations to substrate embeddedness from the current work, results indicate increased sediment supply has degraded water quality, physical habitat, and aquatic refugia in HCW, especially in the agricultural headwaters and the lower urban reaches.

Collectively, previous studies showed streams of the Midwestern USA are commonly associated with increased sediment supply and degraded stream health [13,15,57]. For example, Gellis et al. [57] noted channel sources of sediment accounted for the majority (>50%) of bed sediment in 79% of 99 Midwestern watersheds sampled. Increased channel sediment supply and subsequent bed sedimentation can bury riffle habitat, reduce egg and fry survivorship, and lower prey densities [63,64]. Results from previous studies often focused on agricultural influence on stream bank erosion and sedimentation which was observed in the current work. However, it should be noted that urban land use may cause a greater influence on channel morphology and streambed sedimentation relative to agricultural land use. For example, substrate embeddedness was observed to decrease between Site #1 and #2 where agricultural land use accounted for 55% of total catchment area. Continuing downstream, substrate embeddedness began to increase between Sites #2 and #3 at the rural—urban interface of HCW which was associated with about 7 to 13% urban land use in this study. Thus, results from this study were in agreement with Paul and Meyer [65] who noted urban land use can account for a small percentage of total catchment area while causing a disproportionate influence on water quality and stream health relative to other land uses.

### 4.3. Management Implications and Future Research

The PHA and nested-scale experimental watershed study design presented in the current work is a useful model for managers that need to elucidate casual factors, target critical source areas, and thus, guide regional stream restoration efforts of mixed land use watersheds globally. The intensive field data collection associated with this assessment made it possible to estimate downstream changes in metrics important for understanding available stream physical habitat and biological refugia across an agricultural-urban land use gradient. Assuming that reduction of sediment supply below sediment

transport capacity will help to decrease substrate embeddedness, management efforts might focus on reducing land use alterations to (1) erosive forces of surface runoff and streamflow, and (2) channel incision and bank losses in watersheds prone to increased sediment supply.

A complete decoupling of the simultaneously occuring natural and anthropogenic factors (legacy and ongoing) that influence channel geomorphology and streambed sedimentological characteristics was beyond the scope of the current work. Additionally, results from this study do not show temporal variablity of channel geomorphology and substrate composition. Thus, results may not reflect the total influence of past and present land use alterations that are expected to continue to alter channel geomorphology and substrate composition over long time periods. Future work focused on re-measuring channel geomorphology, substrate frequency and embeddedness may elucidate relationship between land use change and subsequent alterations to stream physical habitat.

## 5. Conclusions

Key findings from the current work point to (1) agricultural and urban land use alterations on channel geomorphology and stream substrate embeddedness, and (2) channel geomorphology as an indicator of stream substrate embeddedness in a mixed land use Midwestern stream. Expected relationships between drainage area and channel geomorphology were altered. Agricultural and urban land use explained nearly all of the variance in average width to depth ratios ($R^2 = 0.960$; $p = 0.020$; n = 5), and maximum bank angle ($R^2 = 0.896$; $p = 0.052$; n = 5). Also observed were reduced frequencies of riffle habitat at Site #1 in the agricultural headwaters (13.1%) and Site #5 (12.2%) in the lower urban reaches. Increased rate of change in percent embeddedness of riffle habitat exceeding 5% km$^{-1}$ in the lower urban reaches indicated a disproportionate influence of urban land use on hydromorphology. Results showed nearly all the variability in channel unit frequency was explained by increased frequency of fine substrate type ($0.927 \leq R^2 \geq 0.854$; $p \leq 0.016$; n = 5). These results highlighted observed streambed siltation in Hinkson Creek, especially in the agricultural dominated headwaters and the lower urban reaches. Thus, results from this study point to a critical need to mitigate observed agricultural and urban land use impacts on stream hydrogeomorphology.

Given the influence of stream physical habitat on stream ecosystem health, stream physical habitat assessments are an integral component of regional stream restoration efforts. The robust observed data set collected for this study can provide critical information needed to guide regional policy development and watershed management efforts. Results of this study are particularly important for regional management efforts considering (1) rigorous hydrogeomorphic data sets are rare, and (2) the magnitude of alterations to channel geomorphology, and streambed composition in Midwestern streams of the USA. Results from this study quantitatively characterize channel geomorphology, substrate frequency, and embeddedness across 56 km stream length in the Midwestern USA where streams are commonly burdened with increased channel sediment supply, increased suspended sediment and total phosphorous, and problems associated with substrate siltation. While results from the current work are regionally applicable, key findings may also be useful to guide policy development and management decisions in physiographically similar watersheds globally.

**Author Contributions:** Conceptualization, J.A.H. and S.J.Z.; Formal Analysis, S.J.Z.; Investigation, J.A.H.; Data Curation, S.J.Z.; Writing-Original Draft Preparation, S.J.Z.; Writing-Review & Editing, J.A.H.; Supervision, J.A.H.; Project Administration, J.A.H.; Funding Acquisition, J.A.H.

**Funding:** This research was funded by the Missouri Department of Conservation and the U.S. Environmental Protection Agency Region 7 through the Missouri Department of Natural Resources (P.N: G08-NPS-17) under Section 319 of the Clean Water Act and through joint agreement of the University of Missouri, the City of Columbia, and Boone County Public works and partners of the Hinkson Creek Collaborative Adaptive Management (CAM) program. Additional funding was provided by the National Science Foundation under Award Number OIA-1458952, the USDA National Institute of Food and Agriculture, Hatch project accession number 1011536, and the West Virginia Agricultural and Forestry Experiment Station. Results presented may not reflect the views of the sponsors and no official endorsement should be inferred.

**Conflicts of Interest:** The authors declare no conflict of interest.

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
