# Peer review of "Characterizing Land Use Impacts on Channel Geomorphology and Streambed Sedimentological Characteristics"

_water, doi:10.3390/w11051088_

Round 1
Reviewer 1 Report
The study presented here is based on tremendous amount of very useful data on river morphology. The authors need to be complemented for that. The objective is to document the influence of land use changes (agricultural to urban) on stream sedimentation. Unfortunately, the potential of these data is not fully realized in the study.
I have an impression that the analysis is not well though through and the authors did not have a good concept of analysis that would support their hypothesis. There are many analytical errors and some of the performed analysis is irrelevant. Best example is focusing on spatial location of the substrate within the channel, which from the perspective of study objective is meaningless.
As for obvious errors:
1) In such watershed study the units of width and depth need to be standardized to watershed area, because as the latter gets bigger the river gets wider and deeper. Hence you can’t compare the width downstream with the upstream section.
2) There are 3 hydromorphic units defined: riffle, pool and glides. I am sure this river has more than that. Furthermore distribution of hydromorphic units is flow dependent, but here it is not determined at which flows were they delineated.
3) Although a large amount of cross sections has been measured in 5 long sites, which were showing a gradient in land use, average measurement for each site was used for hydromorphic characteristics and comparative analysis. The problems with that are:
a. Averages are not very meaning full in biological sense. It is the variability that counts much more.
b. The authors reduce their data pool to 5 points for regression analysis. No wonder that the results are dubious. Using all the data with boxplots for example would be much more telling.
Based on the above analysis the authors make some bold conclusions, which I do not see evidence presented for. It is well demonstrated in the claim that presented work is a physical habitat assessment. There is still a long way between this hydromorphological assessment and physical habitat study, therefore this name is inadequate.
Discussion is too long and more focused on other work than mining the analysis presented. It is also repetitive.
I conclude that the study has a very good potential, but the analysis and the paper have to be substantially revised to give it a proper credit.
More comments in annotated file.

Author Response
Please see attached 'water-488767_Response to Reviewer 1'

Reviewer 2 Report
Review of Characterizing land use impacts on channel geomorphology and streambed sedimentological characteristics: A stream physical habitat assessment
Recommendation: Accept with minor revisions
Overview: This paper describes a detailed dataset derived from sampling a stream quite thoroughly. The authors performed numerous statistical analyses looking for connections between land use and stream characteristics. Ultimately, they concluded that a Physical Habitat Assessment is a useful tool in determining the cause of changes to streams, and that urban and agricultural land use make streams deeper, wider, and the banks more steep.
General suggestions: The authors write well, but frequently find themselves listing outcomes (and findings of papers in the literature review), no doubt a result of the complexity of the subject and substantial data collected. This clouds the ultimate message of the paper, which, I believe, is located in lines 212 to 221. Particularly as Water is an interdisciplinary journal, try to streamline a bit, particularly in the discussion and introduction. It may help just to think of shortening the paper.
Specific suggestions:
Line 48: Reword. Features do not become plants.
Line 202: You probably don’t need all of these examples, as they are in Table 3.
Line 216: Explain “multiple regression results”. Do you mean the separate regressions that showed different things? Or multiple regressions produced the same results?
Line 219: Are you saying that via the literature, drainage area should explain bank height, bankfull depth, etc? But that ag and urban landuse have changed this? Please clarify as this the major conclusion of the paper.
Line 232 and 249: Even though you have Table 2, go ahead and add substrate definitions in Table 4 and 5. Will make the paper much easier to read.
Line 299: Too many “for examples” right next to each other.
Line 351: Typo—“the urban”.
Line 519: You have “Results from this study are among the first to quantitatively characterize channel geomorphology, substrate frequency, and embeddedness across 56 km stream length in the Midwestern USA where streams are commonly burdened with increased channel sediment supply, increased suspended sediment and total phosphorous, and problems associated with substrate siltation.” This sounds like you are emphasizing that you are the first to study at 56 km stream. Clarify so that the reader understands what you mean. Be a bit stronger with your conclusions regarding agricultural/urban land use impacts.
Author Response
Please see attached 'water-488767_Response to Reviewer #2_R(2)_5-10-19'.

Reviewer 3 Report
The general comment:
Authors examined the stigma characterizing land use impacts distance in a stream physical habitat survey. They found that streambed substrate samples of pools increased, substrate embeddedness at agricultural area located in the headwaters and urban area located in the lower reaches compared to rural-urban. This relationship remained after controlling for channel geomorphology, and stream substrate variables, such as width to depth ratios, maximum bank angle, and channel width. They also found that any agricultural-urban land use help elucidate regional stream restoration efforts of mixed-land-use watersheds.
Theoretical issues:
This paper focuses on land use and examines the impact of channel geomorphology and environmental contexts. This topic is timely and important considering the increasing importance of overseas subsidiaries
in physical habitat assessment. The paper is well written and easy to follow. The existing literature is well cited. I also applaud that the authors measured the dependent variable and the independent variables in different time periods, which helps address the causality issue. Below I will discuss my main comments and suggestions, which hopefully can help the authors improve the study.
Empirical issues:
This paper is based upon the assumption that agricultural and urban land use assessment is good for channel geomorphology and addresses the question of what factors can affect stream substrate embeddedness in a mixed-land-use. While I concur that drainage area and channel geomorphology is becoming increasingly important for stream physical habitat assessments, not all relationships need to be explained by increased frequency of fine substrate type . It depends upon the strategic objectives in a particular agricultural dominated headwaters. For some agricultural dominated headwaters, being innovative, is key to success while in others implementing integral component of regional stream restoration as directed may be the most important. In other words, while examining channel geomorphology, one needs to consider the regional policy development role in the restoration system. Below are the major issues that I would encourage you to consider in further improving this manuscript:
1. Observed channel geomorphology is probably the key issue. In lines 183-184 you mentioned that: “One-way ANOVA and Tukey Kramer post-hoc multiple comparison tests were used to test for significant differences (CI=95%, p<0.05) in average channel geomorphology between five HCW sub-basins”. Can you between explain in moredetail what average channel geomorphology is. This is an average value for ex ample Channel width ?
2. The impact of local environmental contexts. The paper examines the effects of local Physical habitat assessment .The logic leading to hypotheses is clear. However, how are the effects of observed streambed substrate frequency and different from embeddedness as examined in the channel morphology metrics ?
3. Do local environmental contexts shape the alterations to channel morphology, or do they directly affect substrate composition? Of course, the paper examines the impact of change in stream substrate composition on geomorphology —which was addressed later. In other words, how does the context of channel geomorphology contribute to our understanding of the relationship between environment and land use? To make stronger contributions rather than replicating previous studies in the context of geomorphology and sedimentological characteristics, the authors needs to more closely integrate the context of alterations to channel morphology into the arguments.
4. The impact of streambed substrate, is unique to indicate and thus has great potential to add new knowledge to our understanding of agricultural and urban land use practices. As noted in my general comment, stream physical habitat, may be a part of a mixed-land-use influence on an intensive. A similar argument can be made regarding the relationship between maximum bank angle and channel width at agricultural Site #1. Therefore, even though we observe significant relationships among them, it does not necessarily mean that there are causal relationships between them.
5. All the measures used in this study have been validated. However, when put together, there are overlaps between the key measures. Specifically, channel morphology metrics were included in the measures. The overlaps, which may partially drive the significant relationships between them, further reinforce my concerns on the theoretical relationships between them. Are you sure that all averages with corresponding letters indicate significant differences ?
6. The authors have tried to address issue by measuring the independent variable and the dependent variable in different time periods. Again, I applaud this effort. But considering organizational inertia in the data analysis, there may be considerable persistence in these dimensions over. What do you think ?
7. It is not clear in the paper whether averaged within five nested sub-basins and channel morphology metrics were measured in study period. If it were measured in both time periods, one solution is to control for the prior level of stream length corresponding each gauging site located or to use the difference in nested gauging sites as the dependent variable in order to better capture the impact of environmental contexts on a physical habitat assessment.
8. The reliability of bank to bank variability of streambed substrate frequency for the environmental and variables is relatively low. If these analyses produce similar results, the concern of the low a percent of a substrate type observed at a bank position may be reduced. Is it because of multicollearnarity between the independent variables?
9. If other independent variables are not included, are channel morphology metrics measured still significant?
10. For example, can you potentially argue that channel unit frequency measured at survey sites itself is suppressed in agricultural headwaters contexts because of structural design, and that some elements of organizational design – let’s say, rural-urban mid-reaches - have attributes that could create a better environment? In which case, you could argue that your watershed characteristics, channel morphology, and substrate embeddedness predictors are really structural design attributes? Overall, I think that quantitatively characterized channel is an important and interesting topic. The authors have clearly devoted a great deal of time and effort in this study. I hope that these comments can help improve this study.
A constructive feedback
Having professed my general enthusiasm for the topic and its importance, I have some concerns that I feel are traceable but require substantive effort.
Channel unit frequency measured at survey sites) is interesting, but appears poorly grounded in a theoretical sense. I agree with you that the anthropogenic factor gives the greater prominence in its activities. What I found as a gap in logic is how this translates into strong correlations between channel unit frequency and frequency of fine substrate type? For instance, if the frequencies of pool, riffle, and glide habitats were not significantly correlated we would anticipate more technological investment in that area? Is it the underlying mechanisms such as policy development and management decisions that provide it access to global resources? Is it power to influence or lead decisions and investments? You will need to be more specific on the causal mechanisms. Anthropogenic factors (legacy and ongoing) seems disjointed in a theoretical framework. I agree with you fully that influence channel geomorphology and streambed sedimentological encourages risk taking and show temporal variablity of channel geomorphology as you rightly acknowledge. I need some scaffolding where the logic connects from the prior hypothesis. As it stands, it feels like channel geomorphology and substrate composition don’t really sit well together. There isn’t an overarching sense of framework that holds them together. My suggestion is for you to consider past and present land use alterations have in common or share. This is only a suggestion. You are much closer to the data / logic and can come up with better arguments. My point is simple – you need a more coherent theoretical logic for the choice of hypotheses/variables. For the environmental context issues, I thought that they fit well with what was theoretically expected. As you can see from my point of view, you may want to more tightly connect with the theoretical logic / causal mechanisms. I appreciated the significant effort in the sampling and the survey data collection – this is not often done, and is commendable. The data analysis though only takes into account the final sample of 526 survey sites. However, this could create a sampling bias and distort your analysis.
Technical and methodologcal issue:
There are two suggestions that I would expect that you consider in improving the paper. First, the sample attrition from explained variance by your analysis between channel morphology metrics and substrate embeddedness from left bank to right bank causes some concern. I wonder if you could do some kind of selection model as a two stage geomorhological model? Do you have access to secondary data on the entire data-population? For example, if you knew whether some spatial complexity of streambed substrate composition had a thalweg while others didn’t, then that could be a good selection measure. I know that this is quite a bit of work, but just simple t-tests for non-response seems a watered down way of addressing this potential issue. Second, the data attrition associated with riffles, glides, and pools at five sub-basins with measures of the dependent variable is a concern. Here, I think more could certainly be done along the lines of a selection model. Were there other variables that were left out from the study? Do you have secondary data sources that could be used here?
Summary of the paper:
The manuscript addresses drivers of sediment transport capacity in agricultural land use in the headwaters, particularly taking into account how attributes of the environmental context and substrate composition influence channel morphology. The topic itself is of great importance, especially because variability in substrate embeddedness was are now encouraging such dispersed rather than centralized total sediment supply to allow controls velocity of streamflow, and sourcing of new opportunities and important for the continuing downstream as wel as stream-floodplain-riparian connectivity.
Author Response
Please see attached 'water-488767_Response to Reviewer 2'

Round 2
Reviewer 3 Report
All questions have been covered to improve this research paper. I appreciate your response in accomplishing your goal of having an expedited reviewing process.
Author Response
Please see attached 'water-488767_Response to Reviewer #3_R(2)_5-10-19'
